# LEARNING PHASE REPRESENTATIONS FOR MICROSTRUCTURAL SEGMENTATION IN METALLOGRAPHIC IMAGES THROUGH EXPERT KNOWLEDGE

## ABSTRACT

Automated segmentation of metallographic images containing multiple phases such as martensite, ferrite, and pearlite is essential for quantifying different phases and thereby helping in the understanding properties of materials. Segmentation of these phases is challenging as they often exhibit overlapping boundaries, similar textures, and other more complexities that require a holistic understanding of the microstructures and correct phase representation within the image. To this end, we propose a novel approach for learning phase representations that captures the subtle differences between phases. Our proposed Phase Learning Module strategically integrates phase ratio information with image encodings to produce ratio-aware features that preserve critical spatial details. Materials scientists can roughly estimate phase ratios by examining an image, and our proposed model leverages this expertise. While we use expert-estimated phase ratios during inference, we train a model using accurate phase ratios obtained from target mask images. To our knowledge, this is the first use of class ratios as input in a deep learning segmentation model that serves as constraints to guide consistent phase proportions in predictions. Experimental results demonstrate segmentation performance improvements on both private and public datasets, with a 5.65% increase in Dice scores on the private dataset and a 6.48% improvement on the MetalDAM dataset with only 1.07% increase in model parameters. Furthermore, visualizations show that our approach leads to learning of more distinct and better phase representations across models. The code and private dataset will be made publicly available.

## 1 INTRODUCTION

Microstructure analysis is a fundamental aspect of materials engineering, without which no scientific understanding of engineering materials can be achieved (Biswas et al., 2023b). Microstructures in material science refers to the arrangement of phases, grains, and defects in a material as observed under a microscope (Yuan et al., 2021). The properties of materials vary widely depending upon the microstructure specifications and underlying phase constituencies (Matthews, 1998). A phase is a part of microstructure that has a distinct crystalline structure and chemical composition (Sanyal et al., 2021). Accurate identification and segmentation of different phases in a microstructure can lead to understanding the characteristics of a material (Martin, 2006).

Metallographic images are obtained using optical microscopy (OM), scanning electron microscopes (SEM), Electron Backscatter Diffraction (EBSD), among others, and are used to identify and characterize different phases in a microstructure (Nogara & Zarrouk, 2018; Gintalas & del Castillo, 2022). While OM provides low-magnification images suitable for approximate assessments, it lacks the resolution needed for fine microstructural features (Zhan et al., 2007). EBSD offers high-resolution imaging and detailed phase information but involves high acquisition costs and requires expert interpretations (Kim et al., 2021; Swain et al., 2023). SEM strikes a balance by providing detailed high resolution images with lower cost and complexity compared to EBSD. However SEM images have difficulty in segmentation due to the high visual similarity between different phases, overlapping boundaries, and complex textures as shown in Fig. 1. These challenges often lead to ambiguities

in phase separation and makes traditional data-driven segmentation models prone to errors (Mollens et al., 2022).

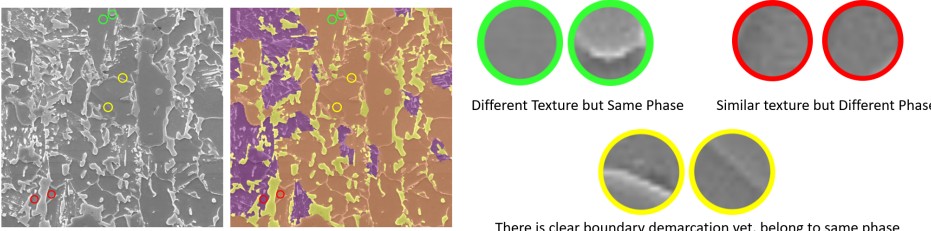

Figure 1: A metallographic image captured by SEM which contains three phases and its overlayed labeled image from private dataset demonstrating the complexity of segmentation.

The deep learning based methods used for material segmentation are mostly based on convolutional neural networks (CNNs) which learns kernel mappings from input to output data without explicitly designing kernels using domain knowledge (Na et al., 2021). Semantic segmentation was popularly used since it could perform phase segmentation by assigning each pixel in the image to one of the pre-defined categories (Santos et al., 2019). DeCost et al. (2019) used PixelNet (Bansal et al., 2017) to find microstructures but was limited only to a certain phase and not multi-phase segmentation. Lai et al. (2009) explored segmentation on contrast and etched steel datasets acquired via SEM and OM imaging. However, it was only optimized to handle images with high contrast variations. Durmaz et al. (2021) proposed the use of U-Net (Ronneberger et al., 2015a) architecture for distinguishing bainite phase regions from irregular ferrite phase and addressed the complex phase problem by framing into binary segmentation task. The work of Luengo et al. (2022) not only included in-depth analysis on complex microstructures by comparing supervised, semi-supervised and unsupervised methods but also proposed a new benchmark MetalDAM dataset for public evaluations. The comparison found out the effectiveness of semantic segmentation in achieving high accuracy compared to other segmentation methods. Recently, Biswas et al. (2023a) performed phase segmentation using a union of attention guided U-Net models by using HSV, RGB and YUV color spaces of input image to capture different characteristics of the phases.

All previous methods have performed segmentation of phases in metallographic images based solely on the visual semantics of the input images. However, it remains unclear whether the models actually learn meaningful phase representations. Do the models develop a holistic understanding of the phases within a metallographic image, or do they merely perform pixel-based classification to accomplish the segmentation task? Fig. 2(a) shows the visualization of phase representations across previously used models. The visualizations are performed from the output of the image encoder and then Principal Component Analysis (PCA) is used to select three most important channels which are then plotted out. It can be seen that the original image embeddings of the models lack in distinguishing different phases in the microstructures, indicating that the models do not fully understand the phase characteristics, despite achieving reasonable segmentation performance.

The use of foundation models like Segment Anything Model (SAM) (Kirillov et al., 2023) demonstrate the potential of incorporating additional inputs such as visual prompts to guide segmentation. These models have shown that conditions or rough hints can significantly enhance segmentation performance. Motivated by this approach, we observed that material scientists can roughly estimate the phase ratio by examining a metallographic image. In this paper, we propose a novel approach of learning the phase representations using the phase ratio as domain knowledge input into the model. We perform adaptations on SAM using LoRA (Hu et al., 2021) which is detailed in the Appendix A.1.

Our method integrates phase ratio information into the neural network architecture through a dedicated ratio encoder. The ratio encoding is then combined with image encodings to produce ratio-enhanced features. These enhanced features are refined further through spatial-aware encodings that preserves the spatial relationships and boundaries between different phases. We also introduce regulators in our model that modulate the influence of domain knowledge and allow the network to perform well even when no ratio input is provided. The phase ratio is calculated from the ground

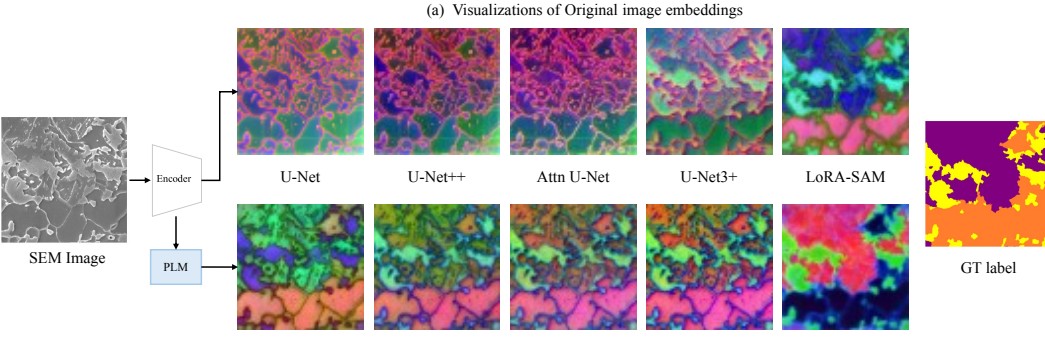

Figure 2: Visualizations of image encodings of various models. (a) is visualization of original image encodings and (b) is visualization of image encodings with proposed Phase Learning Module (PLM). Embeddings with PLM show better distinction and clear representation of phases as compared to original image embeddings. Notably, the LoRA-SAM embeddings using PLM closely resembles the phase distribution of the ground truth segmentation mask (GT label), highlighting the effectiveness of our proposed module. The embeddings are visualized using PCA by selecting top three channels.

truth segmentation masks during training and an estimated ratio is input into the model by the expert after observing the image during inference. Our proposed method accomodates the given user input and adjusts the segmentation outputs by emphasizing or de-emphasizing specific phases. Through this, we demonstrate the effectiveness of our approach and show both quantitatively and qualitatively that the model understands and distinctly represents the phases present in the microstructure. The effectiveness of our approach on phase representation can be visualized in Fig. 2. When the previous models implement our proposed phase learning module, the latent space encodings exhibit far better phase separation and representation, with clearly defined boundaries and less overlap between phases. The visual improvement in phase distinction is a direct result of strategic combination of domain-specific knowledge of phase ratios with image encoding, which allows the network to maintain consistency in phase representation. This enables the model to not only learn the visual semantics of the phases but also understand their inherent structure and proportions. Our proposed method achieves a 5.65% increase in Dice scores on the private dataset and a 6.48% improvement on the MetalDAM dataset on average. The key contributions of the paper are described as follows:

- We introduce a novel methodology for learning phase representations by incorporating domain knowledge in the form of phase ratios. We show both qualitatively and quantitatively the effectiveness of our proposed methodology in learning phase representations across models.

- We strategically merge the image encodings with domain specific information using our proposed Phase Learning Module. To the best of our knowledge, its the first time where class ratios are used as input into a deep learning model and are used as constraints guiding the model to maintain a consistent proportion of each phase in its predictions.

- Our proposed methodology allows experts in the field to input observed or potential phase ratios in the image during inference. Our experimentation shows that there is an increase in performance of the model when the input ratio accuracy during inference is greater than 66.2%.

- Our experimental results further demonstrate the improvement in segmentation accuracy by achieving state of the art results in the publicly available MetalDAM dataset and noticeable performance improvements on private dataset which will be made publicly available along with the source code.

## 2 METHODOLOGY

The overall architecture is shown in 3.The input metallographic images are initially processed by an image encoder which generates an encoded representation of the image. To extract phase-specific

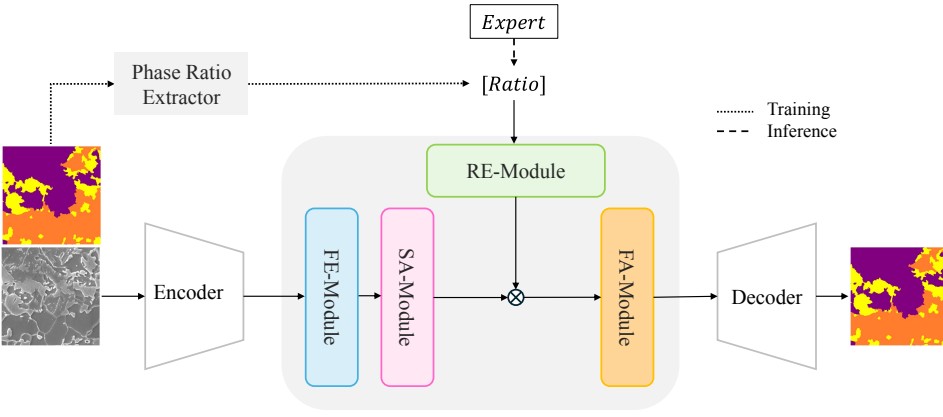

Figure 3: Overview of the proposed method.

information, the encoding is passed through the Feature Extraction (FE) module, where the features corresponding to each phase are segregated. The output for each phase is then further refined by the Spatial Awareness (SA) module, which adds coordinate information to preserve spatial details. During the training phase, the phase ratio—representing the proportion of each phase in the image—is extracted from the ground truth segmentation mask. However, during inference, the phase ratio must be provided by the user- either as an estimated value based on visual inspection or as an approximation when no exact ratio is known. This ratio is represented as a vector of shape $[1 \times n]$, where $n$ is the number of phases. It is processed through the Ratio Encoder (RE) to produce ratio encodings. These are then merged with the image encoding generated by the image encoder via SA module and Feature Aggregator (FA) module. To effectively modulate the integration of domain knowledge (i.e., the phase ratio), we introduce two learnable parameters, $\gamma$ and $\delta$, which control the influence of the ratio-enhanced features.

## 2.1 EXTRACTION OF PHASE RATIO

For each metallographic image, the corresponding ground truth segmentation mask contains $k$ distinct phases, each represented by a unique class label. The goal is to calculate the phase ratio for each phase, which is used as input during the segmentation process. The phase ratio for each phase is derived from the number of pixels in the segmentation mask that belong to that phase. Let the ground truth segmentation mask be denoted as $Y$, with each phase represented by a binary mask $y_i$ for $i = 1, 2, ..., k$, where $k$ is the total number of phases. Each binary mask $y_i$ corresponds to the pixels classified into phase $i$. The total number of pixels in the image is denoted by $N$, and the number of pixels assigned to phase $i$ is $n_i$.

The phase ratio $r_i$ for phase $i$ is calculated as the ratio of pixels in phase $i$ to the total number of pixels in the image:

$$r_i = \frac{n_i}{N}, \quad \text{where} \quad n_i = \sum_{p=1}^{N} 1(Y(p) = i) \tag{1}$$

$1(Y(p) = i)$ is an indicator function that equals 1 if the pixel $p$ belongs to phase $i$, and 0 otherwise. Therefore, for an image with $k$ phases, the set of phase ratios is $R = r_1, r_2, ..., r_k$, where $\sum_{i=1}^{k} r_i = 1$.

During model training, the phase ratio is computed from the labeled data and during inference, the user can either provide an approximate estimate or choose not to provide any ratio information at all. Our proposed approach is robust enough to handle both situations effectively. Even in the absence of phase ratios during inference, the model can still perform well as can be seen in 8. However, when phase ratios are available, they can be used to emphasis certain regions in the segmentation output. For example, if the model struggles to accurately identify a specific phase, the corresponding phase ratio can be adjusted to improve the visibility of potential regions as can be seen in 6.

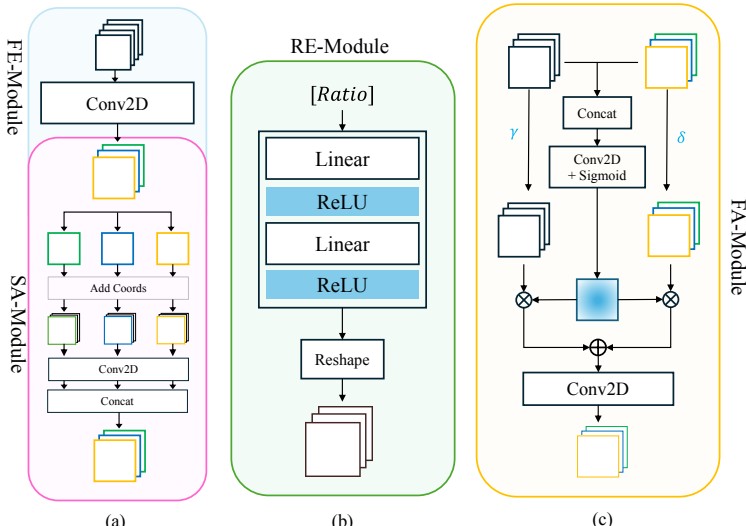

Figure 4: The proposed modules including - Feature Extractor (FE), Spatial Aware (SA), Ratio Encoder (RE) and Feature Aggregator (FA).

## 2.2 IMPLEMENTING PHASE LEARNING

Directly adding the phase ratio information to the image encoding can be ineffective due to the fundamental difference in the dimensionality and representational spaces of these two inputs. While the image encoding captures spatial information in a high-dimensional feature space, the phase ratio is a low-dimensional feature representing global phase proportions. To bridge this gap, we introduce a Ratio Encoder that transforms the phase ratio into a feature representation compatible with the image encoding.

The ratio encoder consists of a 2-layer MLP that encodes the given phase ratio into a ratio encoding having $n$ channels. $n$ denotes the number of phases in the image. The diagram of the ratio encoder is presented in 4(b). After the final segmentation mask is obtained, the phase ratio is calculated again and compared with the phase ratio of the ground truth image. This loss is calculated using Mean Squared Error (MSE) loss Kato & Hotta (2021) and is used to train the ratio encoder to learn the correct phase proportions.

If we were to directly add the phase ratio encoding to the image encoding without any further processing, it might result in a poor integration. We verified this with our experiments, where we observed a decrease of around 2% in segmentation accuracy. This degradation likely occurred because the model lacks context regarding the phase ratio information. The model cannot correlate the phase ratio information with the precise location of the phases in the image encoding which results in a disjoint representation that fails to guide the segmentation process effectively.

In our proposed method, the image encoding is passed through the Feature Extractor (FE) module, which is responsible for segregating relevant feature maps for each corresponding phase. The FE module consists of a convolutional layer that concentrates the image features into $n$ channels, with each channel corresponding to a specific phase ratio. Each of these phase-specific feature maps is then processed using coordinate convolutions Liu et al. (2018) to embed explicit spatial information into the feature maps. This ensures that the model retains spatial context for each phase and allows it to correspond the phase ratio information to the correct phase regions. Finally, the phase-wise feature maps are concatenated and fused with the encoded ratio information from the Ratio Encoder using element-wise multiplication as can be seen in 4(a).

In the final step, the encodings are processed through the Feature Aggregator (FA) Module, which fuses the spatially aware features from the previous stages with the original features generated by the image encoder. This integration is crucial as it not only ensures that the segmentation output is not solely dependent on the phase ratio information but also preserves the original image features.

As a result, the model is able to perform well even in cases where the ratio input is inaccurate or absent.

In FA-module, the spatially aware features and ratio-encoded features are first concatenated channel-wise. These concatenated features are then passed through a convolutional layer with a sigmoid activation function, which helps normalize the feature values and allows for non-linearity in the interaction between the image and ratio features. The output of the convolutional layer is then split into two branches, each multiplied element-wise with the original spatially aware features and ratio-encoded features, respectively. The element-wise multiplication allows the model to modulate the influence of each feature type dynamically. Finally, the two multiplied feature maps are combined through an element-wise addition operation. This fused feature map is then passed through a final convolutional layer to generate the output feature map, which serves as the final segmentation prediction. This is illustrated in Figure 4(c).

This fusion process is regulated by two key parameters: $\gamma$ and $\delta$. These regulators control the influence of Phase Ratio on the final segmentation. A higher value of $\gamma$ reduces the impact of phase ratios. On the other hand, a higher value of $\delta$ increases the influence of the ratio encoder, allowing the model to more heavily rely on the phase ratio guidance.

## 3 EXPERIMENTS AND RESULTS

### 3.1 DATASET

The only publicly available dataset with SEM-based multi-phase micrographs (containing more than two phases) is MetalDAM Luengo et al. (2022). It consists of 42 labeled images across five distinct classes: matrix, austenite, martensite/austenite, precipitates, and defects. In MetalDAM, binary masks were initially used as pre-annotations and were subsequently refined by industry experts. Although this method provided labeled data, the manual refinement introduced some subjectivity and potential inaccuracies into the annotations.

For our experiments, we used MetalDAM dataset along with a private alloy steel microstructure dataset. The private dataset comprises of images captured using SEM and labeled with the assistance of EBSD data through a superlabeler, which provided more objective and detailed annotations. This approach reduces the subjectivity often present in other datasets and ensures more accurate phase labeling. It contains a total of 24 alloy steel microstructure images captured with a SEM at varied magnification levels (2700x magnification - 6 images, 3000x magnification - 10 images, and 5000x magnification - 8 images). The samples have a tensile strength of 780 MPa. Each image includes three types of microstructures or phases: Bainite, Ferrite, and Martensite.

Both datasets (private and MetalDAM) were split into training, validation, and test sets with a 70-20-10 ratio, respectively. To address the limited number of images, we applied sliding window techniques and various geometric and photometric augmentations to increase the effective size of the dataset. These augmentations included flipping, rotation, scaling (magnification), intensity adjustments, gamma correction, and contrast-based transformations.

### 3.2 EXPERIMENTS

Our models were implemented using the PyTorch framework and was run on a single NVIDIA Titan RTX GPU, with an Intel Core i7 6700 CPU, running on the Ubuntu 22.10 operating system. The models were trained with the Adam optimizer Kingma & Ba (2017), with an initial learning rate of $1 \times 10^{-5}$ and a batch size of 8 for 40 epochs. We used Mean Squared Error (MSE) loss to calculate the phase ratio deviation between the ground truth and predicted mask. A weighted sum of Dice Coefficient and Cross-Entropy (Dice CE) loss was considered an appropriate metric to evaluate the performance of the models Naser & Alavi (2023). The $\gamma$ and $\delta$ parameters in the model that are used as regulators for feature aggregator were learned by the model during training. The phase ratio input during inference mimics the expertise of the user having 90% accurateness and was calculated using Appendix A.2.

**Comparison of proposed method performance.** Table 1 presents the performance of various models on steel microstructure segmentation tasks for both the private and MetalDAM datasets. The table

Table 1: Comparison of model performance with and without our proposed Phase Learning Module (PLM) on private and MetalDAM datasets using Dice scores.

| Model | Private Dataset | | MetalDAM Dataset | |
| --- | --- | --- | --- | --- |
| | Baseline | w/ PLM | Baseline | w/ PLM |
| U-Net (Ronneberger et al., 2015b) | 53.85 | 65.39 | 76.43 | 86.33 |
| U-Net++ (Zhou et al., 2018) | 76.25 | 82.12 | 80.82 | 87.04 |
| Attn U-Net (Oktay et al., 2018) | 78.43 | 82.88 | 83.06 | 88.76 |
| U-Net3+ (Huang et al., 2020) | 79.96 | 84.24 | 84.34 | 90.21 |
| nnU-Net (Isensee et al., 2021) | 79.68 | 83.12 | 82.89 | 88.53 |
| TransUNet Chen et al. (2021) | 79.82 | 84.11 | 84.25 | 89.22 |
| UCTransNet (Wang et al., 2022) | 81.46 | 85.29 | 85.57 | 90.88 |
| LoRA-SAM | 84.42 | 88.79 | 86.21 | 92.34 |

| Ground Truth | LoRA-SAM | LoRA-SAM w/ PLM |

Figure 5: Segmentation results of the proposed method with the integration of phase learning module. The top row shows segmentation results for the private dataset, while the bottom row shows results for the MetalDAM dataset. Some of the improved regions are highlighted using the insets in each image.

compares baseline performances of each model with its performance when integrated with our proposed framework, using Dice scores as the evaluation metric. There is a 11.54% improvement in the private dataset and a 9.9% boost in the MetalDAM dataset using U-Net Ronneberger et al. (2015b) model, indicating that the phase ratio guidance provided by proposed method is most beneficial for simpler models. Advanced architectures like U-Net++ Zhou et al. (2018) and Low Rank Adaption-SAM Hu et al. (2021); Kirillov et al. (2023) also show improvements of 5.87% and 4.37%, respectively, in the private dataset, and 6.22% and 6.13% in the MetalDAM dataset. Even high-performing models, such as LoRA-SAM, exhibit consistent improvement, showing that our proposed method can enhance performance across various architectures by incorporating domain-specific information. LoRA-SAM was trained with various ranks out of which rank of 512 performed better. More detailed experiments can be found in the appendix A.1 section.

In Figure 5, we present a qualitative comparison of segmentation results on both the private dataset (top row) and the MetalDAM dataset (bottom row). The results clearly show that the implementation of phase ratio guidance leads to more accurate segmentation. The ratio constraints placed by the ratio encoder force the model to produce segmentation maps that better adhere to the expected phase proportions.

**Comparison between different modules.** Table 2 and Table 3 show the impact of each Ratio Encoder (RE), Spatial Awareness (SA), and Feature Aggregator (FA)—on model performance. When using only the ratio encoder (RE), the performance of all models drops compared to their baseline scores. This degradation likely occurs because the model, when using RE alone, lacks spatial

Table 2: Comparison of model performance on the private dataset with different configurations.

| Model | Baseline | w/ RE | w/ SA | w/ SA+FA | w/ RE+SA | w/ RE+SA+FA |
|---|---|---|---|---|---|---|
| U-Net (Ronneberger et al., 2015b) | 53.85 | 51.24 | 54.03 | 54.08 | 57.37 | 65.39 |
| U-Net++ (Zhou et al., 2018) | 76.25 | 74.61 | 76.35 | 76.39 | 77.92 | 82.12 |
| Attn U-Net (Oktay et al., 2018) | 78.43 | 77.83 | 78.61 | 78.57 | 81.46 | 82.88 |
| U-Net3+ (Huang et al., 2020) | 79.96 | 79.92 | 80.22 | 80.17 | 81.87 | 84.24 |
| nnU-Net (Isensee et al., 2021) | 79.68 | 77.16 | 79.82 | 79.76 | 81.57 | 83.12 |
| LoRA-SAM | 84.42 | 86.60 | 84.53 | 84.56 | 87.31 | 88.79 |

context regarding the phase ratio information. Without proper spatial awareness, the model cannot accurately correlate the phase ratios with the corresponding phase locations in the image encoding and leads to a disjointed representation that fails to effectively guide the segmentation.

However, once the phase ratio is integrated with spatial awareness, the models show significant improvement. This indicates that the spatial information is crucial for effectively using phase ratios in guiding the segmentation process. Furthermore, using only SA does marginally improve the performance but using with RE and FA produces the best results. The qualitative analysis of the effectiveness of the proposed components is shown in Figure 6.

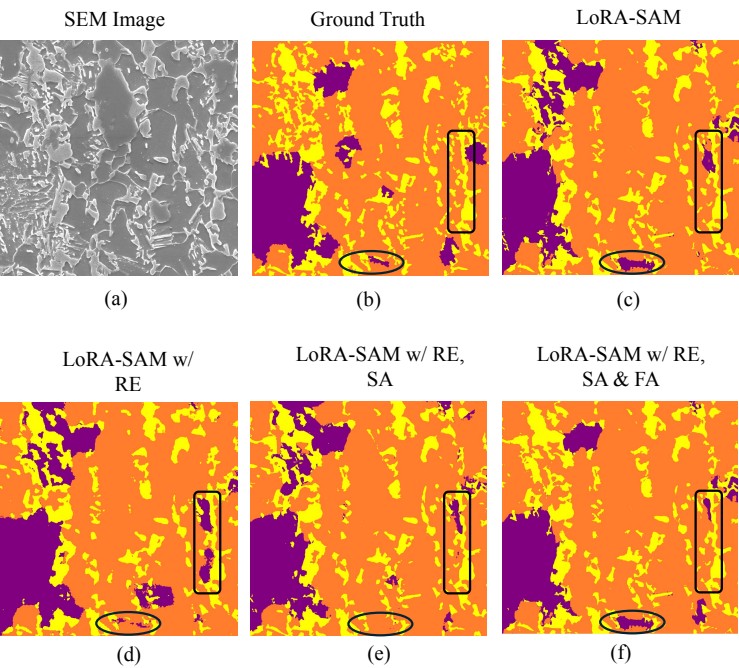

Figure 6: Segmentation results of the models described in Table 2. The colors purple, orange and yellow correspond to phases - Bainite, Ferrite and Martensite respectively.

**Effectiveness of phase ratio during inference.** Figure 7 qualitatively demonstrates the impact of incorporating phase ratio as domain knowledge in the proposed methodology. The top portion of Figure 7 represents the private dataset, where the numbers below the images indicate the phase ratios for the three phases present in the image: martensite, ferrite, and bainite. In Sample Inference 1, the ratio of martensite is increased from 0.15 to 0.25, resulting in the predicted image showing a larger area of martensite which is also highlighted by the green square. Similarly, in Sample Inference 2, the ratio of ferrite is increased, leading to an expanded ferrite region, as indicated by the red circle. In Sample Inference 3, the ratio of bainite is increased, and the model responds accordingly, expanding the bainite region, marked by the blue triangle. The bottom portion of Figure 7 represents

Table 3: Comparison of model performance on the MetalDAM dataset with different configurations.

| Model | Baseline | w/ RE | w/ SA | w/ SA+FA | w/ RE+SA | w/ RE+SA+FA |
|---|---|---|---|---|---|---|
| U-Net (Ronneberger et al., 2015b) | 76.43 | 73.37 | 76.51 | 76.55 | 79.82 | 86.33 |
| U-Net++ (Zhou et al., 2018) | 80.82 | 79.41 | 81.07 | 81.23 | 83.76 | 87.04 |
| Attn U-Net (Oktay et al., 2018) | 83.06 | 81.58 | 83.18 | 83.12 | 85.07 | 88.76 |
| U-Net3+ (Huang et al., 2020) | 84.34 | 84.13 | 84.41 | 84.36 | 87.46 | 90.21 |
| nnU-Net (Isensee et al., 2021) | 82.89 | 81.33 | 82.95 | 82.91 | 86.09 | 88.53 |
| LoRA-SAM | 86.21 | 85.32 | 86.41 | 86.37 | 88.59 | 92.34 |

Figure 7: Qualitative analysis of the effects of our proposed phase ratio guidance. The top row shows a sample image from the private dataset, followed by the ground truth and corresponding inference results. Similarly, the bottom row represents an image from the MetalDAM dataset. The numbers below each image indicate the phase ratios for the corresponding segmentation. Color-matched polygons highlight the changes in phase representation between the ground truth and inference images when corresponding phase ratio is provided.

the MetalDAM dataset. Similarly in sample inference 1 and sample inference 2, the phase ratio was changed and the model tried to accommodate the changes based on phase ratio input that can be observed in the highlighted regions of yellow circle and red box.

## 4 OBSERVATIONS AND LIMITATIONS

Several key observations were made from the experimental results and qualitative analysis, which demonstrate the effectiveness of the proposed methodology. The integration of phase ratio guidance significantly improved the performance of all models across both private and MetalDAM dataset. The Figure 9 shows the impact of input phase ratio guess accuracy on segmentation performance. During the inference the phase ratio that is domain information cannot be obtained and has to input by the user. The user can perform a guess on the phase ratio based on input image observation and our model is able to perform better than baselines if the guessed phase ratio input accuracy is better than 66.2%.

Figure 8 illustrates both the advantage and limitation of our approach. It can be observed that the model performs well in case where no phase ratio input is provided (phase ratio defaulted to 0) but the performance degrades when improper and highly inflated phase ratio input is provided during inference. Such incorrect phase ratio input can negatively impact the model's segmentation

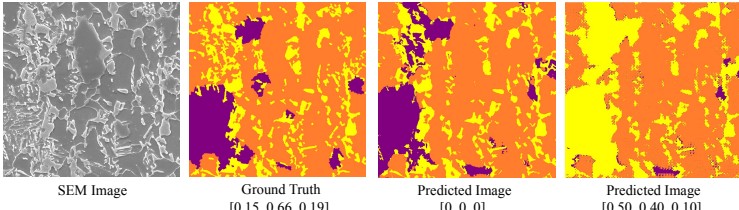

SEM Image      Ground Truth [0.15, 0.66, 0.19]      Predicted Image [0, 0, 0]      Predicted Image [0.50, 0.40, 0.10]

Figure 8: Shows the segmentation result when no phase ratio input is provided and when highly inflated ratios are provided as input during inference.

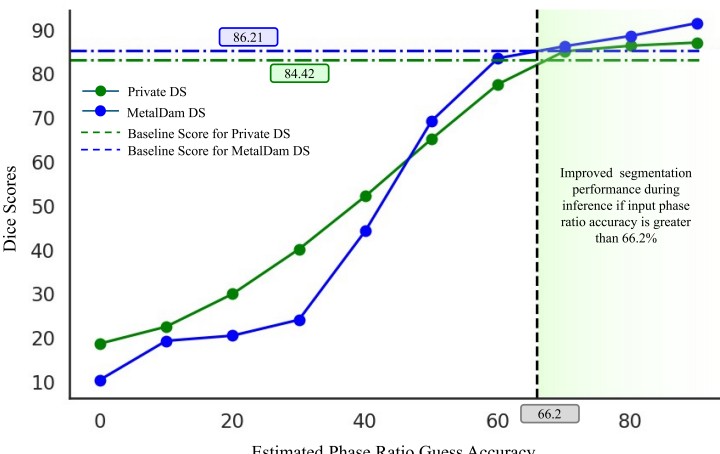

Figure 9: Impact of phase ratio guess accuracy on segmentation performance of our proposed method. The graph illustrates the relationship between accuracy of the phase ratio input during inference and the model's performance. The results shows that segmentation performance improves as the input phase ratio accuracy during inference increases. It surpasses the baseline performance of the model when the guess accuracy of phases during the inference is greater than 66.2%.

performance. This limitation is further confirmed by Figure 9 where significant deviations in phase ratio inputs lead to decreased model accuracy. The results indicate that while the model is robust when phase ratio guesses are reasonably accurate, large variations from the true ratios reduce the effectiveness of proposed method.

## 5 CONCLUSION

In this paper, we proposed a novel method of learning phase representations for microstructural segmentation in metallographic images where we leveraged expert's knowledge on phase ratios to improve segmentation performances. By integrating phase ratio information into the segmentation process, our method provided valuable domain constraints that guided the model to produce more accurate and spatially coherent segmentations. The experimental results on both the private dataset and the MetalDAM dataset demonstrate the effectiveness of our approach, with average improvements of 5.64% and 6.48% in Dice scores, respectively. The primary advantage of our approach lies in its ability to maintain robust performance even when phase ratio information is unavailable during inference. By allowing experts to provide estimates of the phase ratios and the ability of our model to adhere to the same, shows its understanding of the phase constituency in the microstructures. Moreover, our approach improves segmentation accuracy especially when input phase accuracy during inference exceeds 66.2%. While our proposed method of phase ratio guidance demonstrates significant improvements, the need for user-provided phase ratios during inference introduces a potential limitation which will be addressed in future research. Moreover, future research could explore expanding the our proposed method to other domains beyond metallographic images and potentially yielding interesting and broader applications of representation learning.

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

# A APPENDIX

## A.1 ADAPTING SAM FOR MICROSTRUCTURE SEGMENTATION

SAM is built on transformer architecture and has demonstrated remarkable effectiveness in various domains such as natural language processing and image recognition tasks. SAM includes a vision transformer-based image encoder for extracting image features, a prompt encoder for integrating user interactions like bounding boxes, and a mask decoder that generates segmentation results and confidence scores using the image embedding, prompt embedding, and output token. For our approach, we employed the base Vision Transformer (ViT) model as the image encoder. Extensive evaluation indicated that larger ViT models, such as ViT Large and ViT Huge, offered only marginal improvements in accuracy while significantly increasing computational demands. The base ViT model consists of 12 transformer layers, with each block comprising a multi-head self-attention block and a Multi Layer Perceptron (MLP) block incorporating layer normalization.

To adapt SAM for steel microstructure segmentation, we modified the attention layers of the SAM encoder using LoRA. LoRA modifies the attention mechanism in the transformer by introducing low-rank matrices into the query and value computations. The key idea is to decompose the weight updates into two low-rank matrices, which reduces the number of trainable parameters and computational complexity.

For a given weight matrix $W \in \mathbb{R}^{d \times d}$ in the attention mechanism, LoRA decomposes it into $A \in \mathbb{R}^{d \times r}$ and $B \in \mathbb{R}^{r \times d}$, where $r << d$. The modified weight matrix is given by $W' = W + BA$ where $A$ and $B$ are trainable parameters, while $W$ is kept frozen. This decomposition allows the model to efficiently learn task-specific adaptations without extensive retraining. In SAM encoder, each transformer layer's multi-head self-attention mechanism is adapted using LoRA. Specifically, we modify the query ($Q$) and value ($V$) computations as follows:

$$Q' = Q + B_Q A_Q \tag{2}$$

$$V' = V + B_V A_V \tag{3}$$

Table 4: Impact of LoRA Rank on Model Performance across various magnifications in the private dataset.

| Rank | ×2700 | ×3000 | ×5000 | Avg. |
|------|-------|-------|-------|------|
| 256  | 81.22 | 63.91 | 82.58 | 75.91 |
| 512  | 86.02 | 79.43 | 87.80 | 84.42 |
| 1024 | 81.24 | 67.59 | 85.44 | 78.09 |
| 2056 | 85.56 | 79.71 | 87.37 | 84.21 |

Table 5: Performance of SAM model variants for Private dataset. LoRA with rank 512 was chosen for performance comparison.

| Model | Baseline Parameters | Baseline Dice Scores | LoRA Trainable Parameters | LoRA-SAM Dice Score |
|-------|---------------------|----------------------|----------------------------|---------------------|
| SAM ViT-B (base) | 91M | 21.76 | 22.3M | 84.42 |
| SAM Vit-L (large) | 308M | 22.93 | 75.4M | 85.19 |
| SAM Vit-H (huge) | 636M | 23.16 | 156.1M | 86.84 |

To evaluate the impact of different ranks in the LoRA implementation, we conducted experiments using ranks of 256, 512, 1024, and 2056. As shown in Table 4. A rank of 512 provided the best overall performance with balanced computational efficiency and accuracy. While higher ranks had high potential expressive power, they did not consistently improve performance and sometimes led to overfitting or increased computational cost.

Table 5 presents the performance and parameter breakdown of different variants of the SAM model on the private steel microstructure dataset. The table compares the baseline SAM models with their LoRA-adapted counterparts. As the ViT backbone size increases, there is a marginal improvement in baseline Dice scores, ranging from 21.76 (ViT-B) to 23.16 (ViT-H).

Table 6 provides a detailed parameter breakdown for the proposed LoRA-SAM model and its integration with the Phase Learning Module (PLM). The table highlights both the total and trainable parameters for LoRA-SAM and LoRA-SAM+PLM. The base LoRA-SAM model has a total parameter count of 112.0M, with only 22.3M parameters trainable. This efficiency is achieved by adapting the SAM encoder using LoRA, which modifies only specific attention layers while keeping the rest of the model frozen. The addition of the PLM results in only a 1.2 million increase in trainable parameters, which is about 1.07% of the total parameters in the LoRA-SAM model. Despite the small increase in parameters, integrating the PLM leads to substantial improvements in segmentation accuracy, as evidenced by our experimental results.

Table 6: Parameter Breakdown for Proposed Modules

| Model/Module | Total Parameters (M) | Trainable Parameters (M) |
|--------------|----------------------|--------------------------|
| LoRA-SAM | 112.0 | 22.3 |
| LoRA-SAM + PLM | 113.2 | 23.5 |
| PLM (Total) | 1.2 | 1.2 |
| FE Module | 0.0335 | 0.0335 |
| RE Module | 0.0335 | 0.0335 |
| SA Module | 0.0215 | 0.0215 |
| FA Module | 1.179 | 1.179 |

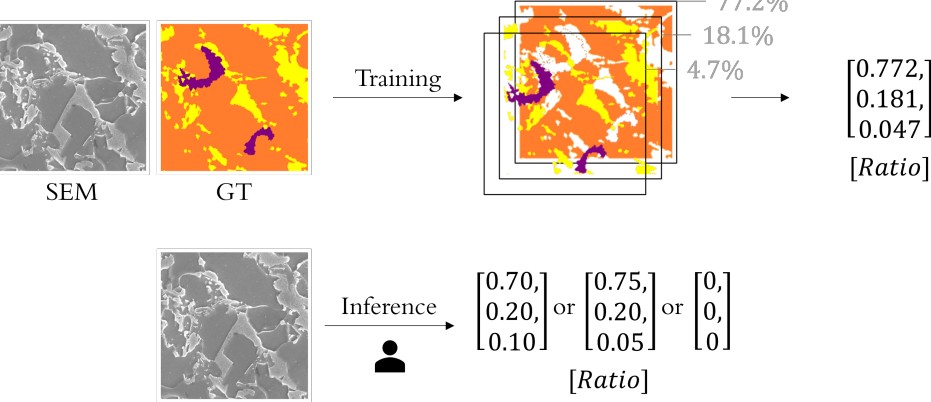

Figure 10: This diagram illustrates the process of obtaining the ratio. During training, the ratio is calculated from the phases of the ground truth image. During inference, an expected ratio is provided as input - which can be a rough estimation of the phases or set to zero.

## A.2 DETERMINING PHASE RATIO

The phase ratio during training is calculated from the ground truth segmentation map as described in Section 2.1 and inference is provided by the expert after observing the metallographic image as shown in Figure 10. The phase ratio input during inference is set to 90% accuracy.

However, here the objective to describe how to determine the phase ratio input during inference with a desired level of accuracy relative to the ground truth phase ratios. For example, if a dataset contains three phases, we may want to evaluate model performance when the guessed phase ratio is 30% off from the true phase ratio. Then for a given phase $P_i$, the guessed phase $G_i$ can be calculated by:

$$G_i = \alpha P_i + (1 - \alpha) \times \delta \qquad (4)$$

where, $\alpha$ represents the desired accuracy, $(1-\alpha)$ is the error or deviation factor and $\delta$ distributes the remaining error across the other phases such that the guessed phase ratios still sum to 1. The value of $\delta$ is calculated as:

$$\delta = \frac{1 - P_i}{n - 1} \qquad (5)$$

It ensures that each guessed phase has the required level of accuracy compared to its true phase proportion while maintaining the sum constraint.

**Example.** Consider a dataset with three phases and the true phase ratios $P_1 = 0.7$, $P_2 = 0.18$ and $P_3 = 0.12$. If the desired phase ratio guess accuracy is 30%, we set $\alpha = 0.3$.

Using the above equations, the guessed phase ratios can be computed as:

$$
\begin{aligned}
P_1' &= 0.3 \times 0.7 + 0.7 \times \frac{1 - 0.7}{2} \\
P_2' &= 0.3 \times 0.18 + 0.7 \times \frac{1 - 0.18}{2} \\
P_3' &= 0.3 \times 0.12 + 0.7 \times \frac{1 - 0.12}{2}
\end{aligned}
\qquad (6)
$$

Guessed phase ratio accuracy required for 30% accuracy would be - $[0.315, 0.314, 0.344]$, for 50% accuracy it would be $[0.425, 0.295, 0.28]$ and for 90% accuracy it would be $[0.6454, 0.203, 0.152]$. This method provides a systematic approach for determining phase ratio inputs during inference with varying levels of accuracy, allowing us to analyze model performance under different levels of deviation from the true phase ratios.

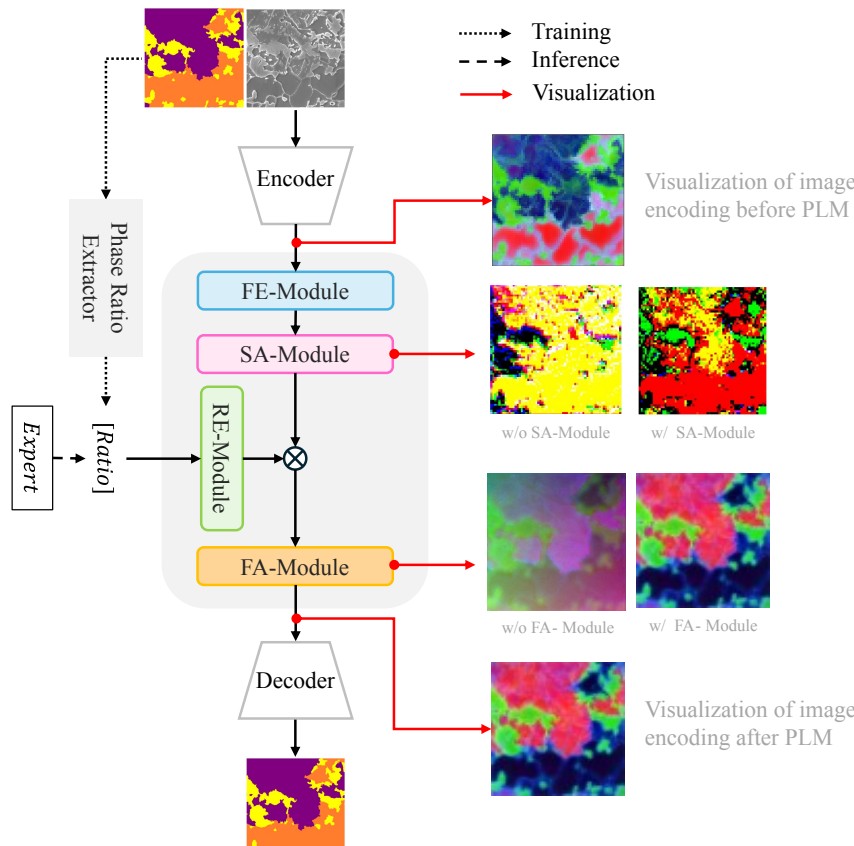

Figure 11: Visualizations of image encodings before and after PLM, with and without SA module, and with and without FA module.

### A.3 VISUALIZATION OF PROPOSED COMPONENTS

Figure 11 provides visualizations of the image embeddings at various stages: before and after applying our proposed method, as well as with and without the Spatial Awareness (SA) and Feature Aggregation (FA) modules. From the figure, we can see that the SA module enhances spatial relationships between the phases, especially when observing the boundary areas. The FA module further improves the encoding by aggregating both the image encodings and ratio encodings. This ensures that the resulting embeddings closely align with the phase proportions seen in the ground truth mask, leading to better-defined phase regions in the output. The FA module ensures that the model captures the correct phase distributions and avoids the blending of similar-looking regions. Comparing the visualizations before and after applying our proposed PLM, we observe a stark improvement in phase representation. The image embeddings after PLM appear significantly more structured and aligned with the actual phase boundaries in the ground truth mask. The distinctions between phases are clearer and more precise, indicating that the PLM effectively integrates spatial and ratio information into the segmentation process. These visualizations were generated using the LoRA-SAM model with a rank of 512, demonstrating how each component of our architecture contributes to progressively refining the phase representations.

### A.4 ANALYSIS ON GAMMA AND DELTA PARAMETERS

The gamma ($\gamma$) and delta ($\delta$) parameters in the Feature Aggregator (FA) module are crucial in determining the balance between the original image features and the ratio-enhanced features generated by the Phase Learning Module (PLM). These parameters dynamically adjust throughout the training process to optimize the contribution of both feature types.

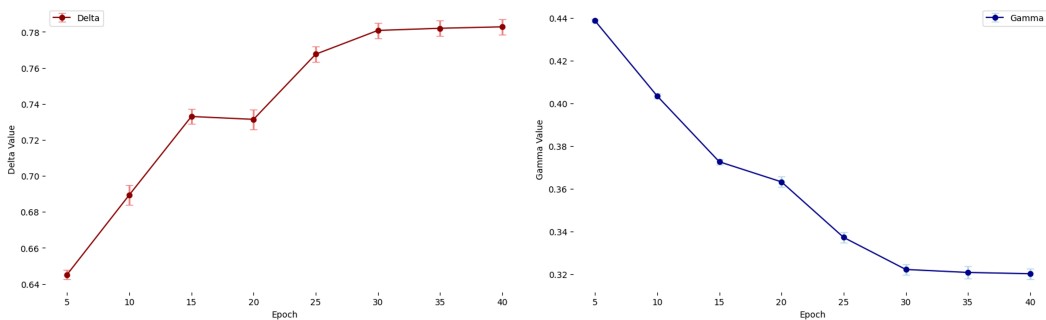

Figure 12: Visualizations of delta and gamma parameter values per epoch. The visualization shows the mean value per epoch with marginal deviation. The parameters stabilize after epoch 30, highlighting the model's convergence in balancing original and PLM-enhanced features.

Figure 12 visualizes the evolution of $\gamma$ and $\delta$ values across epochs, accompanied by their marginal deviations. Both parameters are initialized at 0.50, signifying equal importance for the two feature types at the beginning of training. As training progresses, $\delta$ exhibits a consistent upward trend, reaching a mean value of approximately 0.78 by epoch 40. In contrast, $\gamma$ shows a steady decline and stabilizes around 0.32 by the end of training. This contrasting behavior illustrates the model's increasing reliance on ratio-enhanced features, governed by $\delta$, while progressively reducing emphasis on the original image features, controlled by $\gamma$.

Table 7 complements this visualization by presenting the mean and variance values of $\gamma$ and $\delta$ at intervals of 5 epochs. The low variance for both parameters indicates stable updates and convergence, especially after epoch 30. This stability highlights the model's ability to strike an effective balance between the two feature sets, guided by the adaptive learning mechanism of the PLM.

The final convergence of $\gamma$ and $\delta$ at 0.32 and 0.78, respectively, indicates that the model places significantly more weight on ratio-enhanced features while still retaining a portion of the original image features to maintain contextual information. This helps the model in providing fairly competitive segmentation performance during inference when no phase ratio is provided.

Table 7: Mean values of $\delta$ and $\gamma$ across epochs. Both $\delta$ and $\gamma$ were initialized to 0.5 and then were updated by the PLM model for LoRA-SAM model.

| Epoch | $\delta$ (Mean ± Variance) | $\gamma$ (Mean ± Variance) |
|---|---|---|
| 5 | 0.647 ± 0.003 | 0.439 ± 0.001 |
| 10 | 0.693 ± 0.007 | 0.404 ± 0.002 |
| 15 | 0.735 ± 0.006 | 0.373 ± 0.002 |
| 20 | 0.732 ± 0.008 | 0.364 ± 0.004 |
| 25 | 0.770 ± 0.005 | 0.338 ± 0.004 |
| 30 | 0.785 ± 0.002 | 0.323 ± 0.004 |
| 35 | 0.786 ± 0.002 | 0.321 ± 0.005 |
| 40 | 0.787 ± 0.002 | 0.321 ± 0.004 |

## A.5 QUALITATIVE COMPARISON ACROSS MODELS

Figure 13 shows a qualitative performance comparison of SEM image with and without the proposed PLM across U-Net, nnU-Net, U-Net3+ and LoRA-SAM.

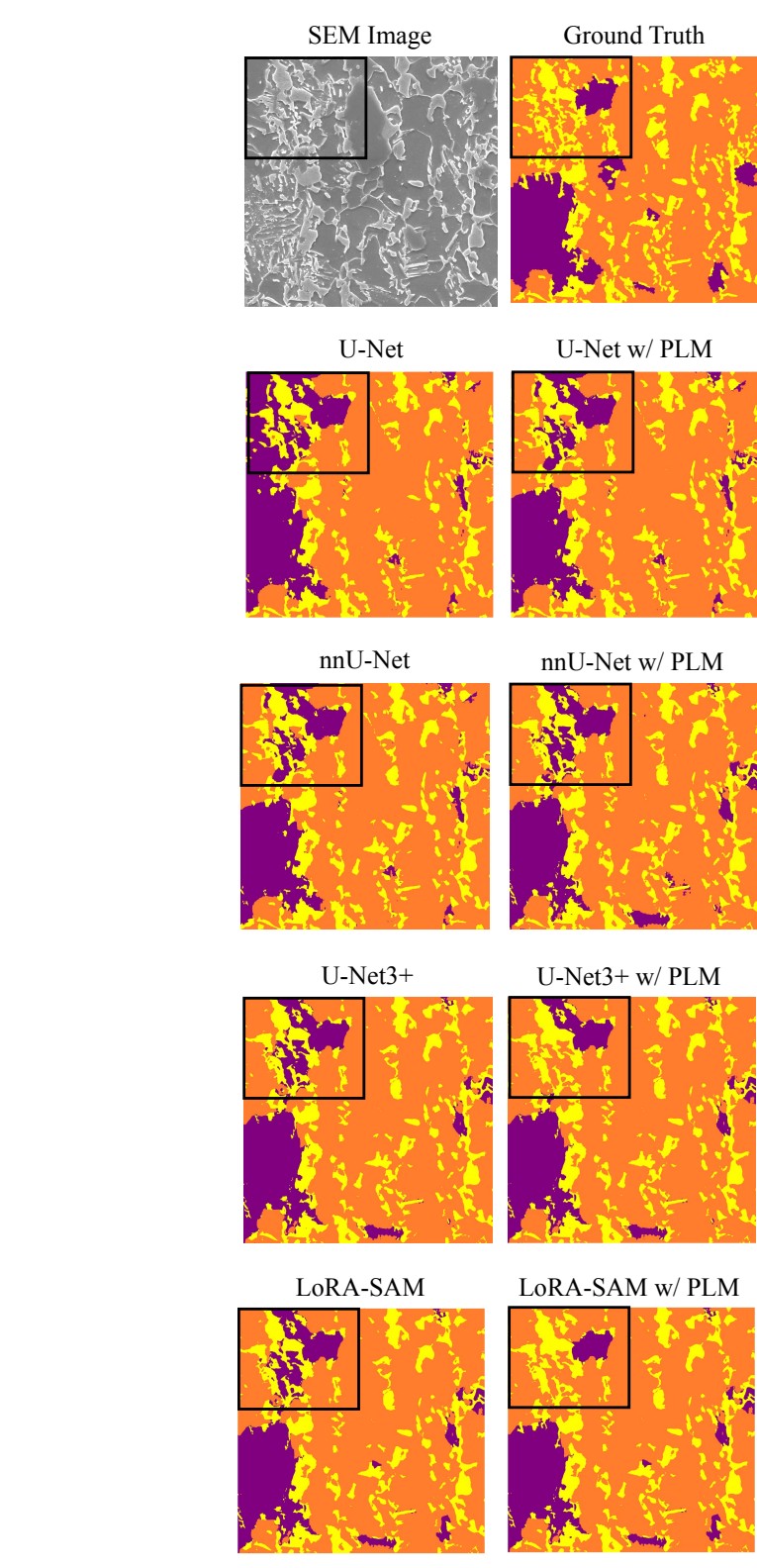

Figure 13: Segmentation results of with and without the Phase learning Module (PLM) across various models with estimated 90% phase ratio accuracy.

