# OpenReview forum: "Learning Phase Representations for Microstructural Segmentation in Metallographic Images through Expert Knowledge"
_ICLR.cc/2025/Conference — ICLR 2025 Conference Withdrawn Submission_

### Official Review · Reviewer_J4Re · 2024-10-16

**Soundness:** 3
**Presentation:** 4
**Contribution:** 3
**Rating:** 8
**Confidence:** 4

**Summary:**

This work presents a module that incorporates domain-specific knowledge to guide a segmentation model, to accurately segment metallographic images. This guidance is the ratio of each segment in the image, while during training it is computed using the GT and during inference it is provided by the operating experts.

**Strengths:**

Problem formulation and motivation is presented really well.
The paper is easy to follow.
It makes sense to utilize a segmentation foundation model and inject domain-specific hints.
It shows pretty consistent (and not negligible) improvements on several segmentation models and on two datasets.

**Weaknesses:**

1) The presented scientific background is too short. I suggest presenting a broader related work section that is separate from the introduction section. A bit more information on what is done on the vision-metallography domains may be helpful, and a bit more information at least on LoRA-SAM as your reported baseline utilizes it. At least - introduce its main components, since you use them in your encoder and decoder.

2) Currently the setting requires the operators to work "harder" as it demands their guidance.
I would suggest to to train another module that will predict the ratio from the input image. Instead of simply calculating it from the GT, predict it from the input and penalize using the GT ratio. This will give you the option to operate using only the image In inference.
It will be interesting to see in this zero-expert-intervention setting, how well does the model perform.

3) It will be interesting to see an analysis of gamma and delta. What did the model preferred to focus on?

Technical issues:
Line 24: "model" -> "a model"

Figure 3: Why is there an arrow from the input to the Phase Ratio Extractor in training? Shouldn't the arrow start from the GT?

Figure 4: Fix the squares behind the yellow square below add coords

Line 247: "denote" -> "denotes"

**Questions:**

The definition of n and k are not clear to me. Is n defined by the total number of phases in the dataset? Something else?
k is identical in each segmentation mask? If not, denote that each mask has a different k^i or something like it.

---

> ### Author Response · Authors · 2024-11-21
>
> We sincerely thank the reviewer for your positive assessment of our work and for the valuable suggestions to improve our paper.  We address your comments and questions below.
>
> 1. **Expansion of the Scientific Background**
> Thank you for this suggestion. We agree that expanding the scientific background and providing more context would strengthen the paper but due to lack of space we have added it in the Appendix A.1 section.
> 2. **Reducing User Effort by Predicting Phase Ratios**
>
>     In the current system the expert has to provide phase-ratio inputs during inference which be a bottleneck. To mitigate this dependency, we are considering a two-stage inference approach in future work, as suggested by Reviewer 4eVt. In the first stage, the model would perform segmentation without any phase ratio input, allowing it to generate an initial prediction based solely on image features. In the second stage, the estimated phase ratios from the initial prediction would be used to refine the segmentation results. This iterative process could enhance performance without requiring precise expert input.
>
> 3.  **Analysis of Gamma and Delta Parameters**
>
>     Thank you for pointing this out. We have conducted an analysis of the gamma (γ) and delta (δ) parameters, which control the influence of the ratio-enhanced features in the Feature Aggregator (FA) module. Our findings are as follows:
>
>     - Through experiments, we found that the model converged to values of **γ = 0.32** and **δ = 0.78** on average. This indicates that the model prefers to place a higher emphasis on the ratio-enhanced features (influenced by δ) while still considering the original image features (controlled by γ).
>     - However, we are also keen on conducting further study on the analysis of Gamma and Delta parameters which we will be conducting in a future study.
>
> We have corrected all the grammatical errors as pointed out by the reviewer. For the clarification of $n$ and $k$,
>
> $k$: Represents the total number of distinct phases in the dataset. It is the number of unique classes or labels used to segment the images into different material phases or regions.
>
> $i$: Identifies each individual phase within the total of $k$ phases. It ranges from 1 to $k$. For example, if $k=3$ , then $i$ can be 1, 2, or 3, corresponding to Phase 1, Phase 2, and Phase 3, respectively.
>
> $n$: Number of pixels and $n_i$ **r**efers to the number of pixels belonging to phase $i$ in a particular image.
>
> $r_i$: Represents the ratio of $n_i$ to the total number of pixels in the image, i.e., the proportion of the image occupied by phase $i$.

---

> > ### Comment · Reviewer_J4Re · 2024-11-21
> >
> > Thank you for your clarifications.
> > Please consider including your analysis and findings of the gamma and delta parameters.
> > This would provide additional insight into the model's operation and clarify, even if slightly, how it works.

---

> > > ### Author Response · Authors · 2024-11-26
> > >
> > > We thank the reviewer for the valuable feedback. We have now added a new section in appendix about the analysis of $\delta $ and $\gamma$ in A.4. The section reports the mean values of the parameters learnt by the model across various epochs. Our analysis indicates that the model places significantly more weight on ratio-enhanced features while still retaining a portion of the original image features to maintain contextual information.

---

### Official Review · Reviewer_4eVt · 2024-10-28

**Soundness:** 2
**Presentation:** 3
**Contribution:** 3
**Rating:** 6
**Confidence:** 4

**Summary:**

This paper proposes a novel method for learning phase representations in the context of metallographic segmentation, effectively capturing subtle differences between phases. The phase learning module introduced in the paper adaptively integrates phase ratio information with image encoding to generate scale-aware features that preserve critical spatial details. During inference, phase ratios can be coarsely estimated from the image to achieve improved segmentation performance. The paper is clearly articulated and well-written.

**Strengths:**

1. The background, motivation, and proposed method are introduced clearly.
2. The comparison with CNN-based segmentation methods is comprehensive.
3. The experimental analysis and explanatory figures are well-presented, and the proposed learnable phase representation method demonstrates a significant improvement in results.

**Weaknesses:**

This paper introduces a learnable phase representation by incorporating phase ratios, statistically derived from ground truth, into the network. During testing, the method relies on expert-estimated phase ratios as conditions, yielding notable performance improvements over the baseline segmentation. However, certain aspects concerning innovation and fairness in comparison could be improved.

Firstly, the use of ground-truth statistical information was previously employed in [1], where such statistical information was constrained within the loss function, thus avoiding the need for conditional input during inference.
[1] Do we really need dice? The hidden region-size biases of segmentation losses.

Secondly, as the approach requires expert-estimated phase ratios during inference, it falls into the category of interactive methods, necessitating a fair comparison with interactive approaches in terms of interaction time and final performance.

**Questions:**

1. I am skeptical about the claim that precise phase ratios are needed during training but that inference can achieve high performance without accurate or even any phase ratio. If the authors' claim holds true, a cascade inference approach could be employed: the first step would involve prediction without phase ratios, followed by phase ratio estimation from the predicted results for a second inference step, thereby potentially removing the need for expert-provided phase ratio assessments.

---

> ### Author Response · Authors · 2024-11-21
>
> We sincerely thank the reviewer for the thoughtful review and  insights that have greatly have helped us identify areas for improvement.
>
> 1. **Comparison with using statistical information in loss function.**
> Thank you for bringing this important reference to our attention. We agree that incorporating statistical information into the loss function is a valuable approach for addressing class imbalance and improving segmentation performance. However, our method  integrates phase ratio information directly into the model architecture. This allows our model to dynamically adjust its predictions based on phase ratios provided during inference. By incorporating expert-estimated phase ratios during inference, our method leverages domain knowledge to improve segmentation accuracy, especially in challenging cases where visual differences between phases are subtle. This real-time adjustment is particularly useful in metallographic analysis, where phase distributions can vary significantly between samples. Our experiments show that the user needs to input  67% or higher accurate phase ratios to have better segmentation result than the baseline model.
> 2.  **Comparison with Interactive Methods**
>
>     You raise an excellent point about the necessity of comparing our method with existing interactive segmentation approaches. However, presently there are not any pre-existing models that allow interactive segmentation of metallographic images. These images present with complex visual semantics that cannot be achieved by model trained on natural images. But we will be evaluating for such in future related researches.
>
> 3.  **Cascade Inference Approach**
>
>     Thank you for this insightful suggestion. We understand your skepticism and agree that exploring a cascade inference approach could be valuable. In our experiments, we observed that the model performs better when no phase ratio input is provided than when incorrect phase ratios are supplied, indicating robustness to the absence of this information.  In the first stage, the model would perform segmentation without any phase ratio input, allowing it to generate an initial prediction based solely on image features. In the second stage, the estimated phase ratios from the initial prediction would be used to refine the segmentation results. This iterative process could enhance performance without requiring precise expert input, thereby improving the model's usability in automated systems or settings with limited domain expertise. We would definitely pursue this in future research.
>
>     Additionally we have added expanded comparisons with more models in Table 1 and have expanded the ablation experiments in Table 2 and 3 that show the importance of each of the proposed module and how ratio information is effectively added using proposed modules. We also include Table 5 and 6 that show the performance comparison across SAM models and parameter breakdown of each of the proposed components. Phase Learning Module adds 1.2M parameters to the existing models and improves performance by over 5% on private dataset and over 6% on MetalDam dataset.

---

> > ### Comment · Reviewer_4eVt · 2024-11-22
> >
> > Thank you for your explanation. However, the phase ratio here is merely a scalar value serving as an input condition, which represents a relatively weak form of supervision and is limited to a single numerical value. On one hand, such a reasoning process likely imposes stringent requirements on the phase ratio, leading to weaker generalization capabilities. On the other hand, its contribution to the overall results appears to be relatively limited. I would appreciate a stronger justification and more robust evidence to support this approach.

---

> > > ### Author Response · Authors · 2024-11-26
> > >
> > > We thank the reviewer for their valuable feedback and appreciate for giving the opportunity to further clarify our methodology and address the raised concerns.
> > >
> > > - We understand your concern and agree that using the phase ratio as a scalar input might seem like a weak form of supervision due to its simplicity as a single numerical value. However, in our method, we do not directly use the phase ratio as a scalar with the image encoding. Instead, we employ a **Ratio Encoder (RE) module** that transforms the phase ratios into multi-channel ratio maps.
> > >
> > >      Our **Feature Extraction (FE) module** separates the original image encoding into $n$-channel feature encodings, where $n$ is the number of phases. We then enhance these features spatially using the **Spatial Awareness (SA) module.**
> > >
> > >      The $n$-channel ratio maps from the Ratio Encoder are merged with the spatially enhanced image encodings. Each ratio map corresponds to a specific phase and is combined with its corresponding phase's visual features. This ensures that the phase ratio information is effectively integrated with the image features for each phase.
> > >
> > >     It can be thought like we add weights into the visual representation of phases based on their ratio. When the phase ratios are changed during inference, the corresponding weights for the phases are adjusted. It allows the model to give more attention to phases with higher ratios and less attention to those with lower ratios and helps in aligning the segmentation output with the expected phase composition.
> > >
> > > - Our integration of **Feature Aggregation (FA) module** ensures that the model does not rely strictly on phase ratios. The FA module uses learnable parameters $\gamma$ and $\delta$ to balance the contributions of the original image encoding and the PLM-enhanced encoding . This way we ensure that the mode retains its original visual representations that is crucial when phase ratio information is absent. While highly inaccurate phase ratios can affect the model's performance by misguiding the attention weights. We also added an analysis on the  $\gamma$ and $\delta$ in appendix section A.4. Our experiments show that the model begins to outperform the baseline when the input phase ratios are approximately **62% accurate or better**.
> > > - Our experiments demonstrate that integrating the phase ratio information leads to notable improvements in segmentation accuracy over the baseline models. We conducted ablation studies (as shown in Tables 2 and 3) to assess the impact of each module.  We also show that our model uses very less number of trainable parameters (Table 6) and is able to learn and put weights on the phases based on the input ratio. We also visualize the effects of each of the module in Figure 11. The results confirm that the PLM learns the phase representation well and plays a crucial role in enhancing performance by using expert’s knowledge of materials.

---

### Official Review · Reviewer_JHKH · 2024-10-30

**Soundness:** 3
**Presentation:** 3
**Contribution:** 2
**Rating:** 5
**Confidence:** 3

**Summary:**

This paper proposes an approach for segmenting metallographic images by integrating expert knowledge through phase ratios, which are estimated by domain specialists. The proposed Phase Learning Module (PLM) enhances the segmentation model’s accuracy by refining image encoding with ratio-aware features, achieving improved performance on both public and private datasets.

**Strengths:**

By incorporating expert phase ratio input, the model bridges domain knowledge with deep learning, improving interpretability and alignment with real-world observations.

The model demonstrates clear performance improvements in Dice scores, achieving substantial segmentation accuracy increases on challenging microstructural datasets.

The model allows input of phase ratios during inference, improving usability in applications requiring expert oversight.

**Weaknesses:**

With only 42 images in MetalDAM and 24 in the private dataset, the training data is limited, potentially impacting the model’s ability to generalize across diverse materials.

The model’s effectiveness relies on accurate phase ratios, which may limit its utility when expert estimations are unavailable or imprecise. Inaccurate phase ratios significantly reduce model performance, which might affect usability in automated systems or those lacking domain expertise.

**Questions:**

How does the model perform in the absence of accurate phase ratio inputs, and are there plans to mitigate this dependency?

Have you considered expanding the dataset, or are there augmentation techniques that could address the limited training data?

Could you provide ablation studies to assess the impact of individual modules, like Phase Ratio integration, SA, and FA, on overall performance?

---

> ### Author Response · Authors · 2024-11-21
>
> We sincerely thank the reviewer for their thoughtful review and for recognizing the strengths of our work. We appreciate the constructive feedback and have addressed the concerns as follows:
>
> 1. **Model Performance Without Accurate Phase Ratio Inputs**
>
>     We acknowledge the importance of evaluating the model's performance in the absence of accurate phase ratio inputs. In our experiments, we observed that the model performs better when **no phase ratio input is provided** than when incorrect phase ratios are supplied (Section 4). This suggests that the model is robust to the absence of phase ratio information but can be adversely affected by inaccurate inputs.
>     To mitigate this dependency, we are considering a two-stage inference approach in future work, as suggested by Reviewer 4eVt. In the first stage, the model would perform segmentation without any phase ratio input, allowing it to generate an initial prediction based solely on image features. In the second stage, the estimated phase ratios from the initial prediction would be used to refine the segmentation results. This iterative process could enhance performance without requiring precise expert input.
>
> 2.  **Limited Training Data and Dataset Augmentation**
>
>     We agree that the dataset size is a limitation. To address this, we have performed extensive data augmentation to enhance the diversity and size of our training data. Our private dataset consists of high-resolution images with dimensions approximately 1660×1640 pixels. We employed various augmentation strategies, including  sliding window augmentation, flipping, rotation, intensity adjustments, gamma correction, contrast variations, and the addition of simplex noise. These techniques expanded our private training set to approximately **5,600 image patches** of 512x512. Similarly, for the MetalDAM dataset, which originally contains images of 1024×768 pixels, we applied the same augmentation methods. This resulted in a training set of about **7,800 images** of 512×512 pixels.
>
> 3. **Ablation Studies on Individual Modules**
>
>     Thank you for this valuable suggestion. We have expanded our experiments to include comprehensive ablation studies assessing the impact of each component of our proposed method. In the revised manuscript, **Tables 3 and 4** now present performance metrics for the following configurations - baseline, w/ RE, w/ SA, w/ SA+FA, w/ RE+SA and w/ RE+SA+FA.
>
>
> Our findings indicate that adding just the RE does not significantly improve performance because the model cannot effectively distinguish between different phases or understand their relationship with the ratios without spatial context. However, incorporating both RE and SA modules leads to increased accuracy, as the model gains information about the phases and their spatial distribution within the image. The FA module is essential for enabling the model to perform well even when no phase ratio is provided; it merges the original image features with the ratio-enhanced features, ensuring robust performance.
>
> Additionally, we have included more performance metrics in **Tables 5 and 6**, where we compare models across different configurations of the Segment Anything Model (SAM) and provide a detailed parameter breakdown of each module in our proposed method. This analysis helps demonstrate the effectiveness and efficiency of our approach.

---

### Official Review · Reviewer_5dQM · 2024-10-30

**Soundness:** 2
**Presentation:** 3
**Contribution:** 1
**Rating:** 3
**Confidence:** 3

**Summary:**

The author builds upon the existing SAM model and designs a new information fusion component called the Phase Learning Module. This module integrates additional information, such as phase ratio data, with image encodings to generate ratio-aware features that enhance segmentation performance. The author tested the model on both private and public datasets, achieving promising results. The application of artificial intelligence to explore less mature fields is commendable, and the author's commitment to making the code and datasets publicly available is beneficial to the field. However, the technical contribution of this work is insufficient for an ICLR paper and may be more suitable for a domain-specific journal.

**Strengths:**

1. According to the author, this is the first instance of using class ratios as input in a deep learning segmentation model, where they serve as constraints to guide consistent phase proportions in predictions.
2. The author claims that releasing the dataset and code may be meaningful for advancing research in a relatively new materials field.

**Weaknesses:**

1. The main issue with this paper is the limited technical contribution. As an ICLR paper, the primary focus should be on the machine learning contribution, but this work mainly relies on fine-tuning SAM. While some technical designs, such as the phase ratio prompt, are introduced, these are clearly minor modifications of SAM, and there are already numerous similar methods. With proper revisions, this could potentially be a good AI for Science paper; however, the current technical contribution is not substantial enough for this problem, and it lacks significant insights for the ICLR audience. It may be more suitable for a domain-specific journal.

2. Although the author claims that the Phase Learning Module is a technical contribution, there are some issues with its design and evaluation. The attempt to integrate external information into the SAM-based segmentation model for improved performance is commendable. However, the practical rationale and potential costs of this approach need thorough evaluation. During training, prompt information is derived from labels, but at the inference stage, acquiring additional information incurs costs. It is important to assess whether this additional cost is justified in real-world applications. Moreover, if the information is obtained from test labels, there is a risk of data leakage.

3. Specific comparison and evaluation shortcomings:

    a. Fairness of Comparison: The author mainly compares basic segmentation models, but even these comparisons lack comprehensiveness. For example, segmentation models based on fundamental transformer architectures, such as TransUNet and UCTransNet, are not sufficiently evaluated. Furthermore, the model with external information is only compared against SAM, ignoring other deep learning models that focus on similar multimodal information fusion. This raises concerns about whether SAM’s framework is necessary for multimodal fusion or if a simpler attention mechanism could achieve similar results.

    b. Metric Selection: The author evaluates the segmentation model using only the Dice coefficient, which may not be sufficient or reliable. Other metrics, such as IoU, NSD, or those assessing boundary accuracy, could provide a more comprehensive evaluation. Additionally, the dataset size is not clearly discussed. If the dataset is small, cross-validation should be performed, and the mean and variance reported, along with statistical tests like a t-test to prove the effectiveness of the newly added module.

    c. Parameter Comparison: Adding the new module likely increases the number of parameters. Comparing only performance without considering parameter count is not entirely fair. Moreover, the author does not compare different configurations of the SAM model (e.g., small, base, large versions), which should be addressed.

**Questions:**

Refer to the discussion in the Weaknesses section. The experimental comparisons need to be more comprehensive, with a wider selection of evaluation metrics and more baselines (including both basic segmentation models and multimodal fusion models, not just SAM). Additionally, a detailed comparison of model parameters and inference speed is necessary. Extra manual experiments may also be needed to evaluate the practicality of acquiring prompt information.

---

> ### Author Response · Authors · 2024-11-21
>
> We sincerely thank the reviewer for their thorough evaluation of our work and for the valuable feedbacks. We address concerns below -
>
> **1. Technical Contribution**
>
> We acknowledge the reviewer's concern regarding the technical contribution of our work in the context of ICLR. While our approach builds upon existing models like SAM, we believe that our proposed Phase Learning Module (PLM) introduces a novel methodology by integrating class ratios as input into deep learning segmentation models. This integration serves as a constraint to guide the model towards consistent phase proportions in its predictions, which is valuable in scenarios where classes in images are difficult to distinguish based solely on visual features.
>
> Our method allows the model to adjust segmentation outputs based on desired class ratio inputs and enhance performance in cases with unbalanced classes or indescribable images. This capability is not commonly found in existing segmentation models and represents a meaningful advancement that could benefit other domains facing similar challenges.
>
> **2. Practical Rationale and Costs**
>
> Our proposed method is designed to assist domain experts who often rely on techniques like Electron Backscatter Diffraction (EBSD) for accurate labeling, which can be costly and time-consuming. By enabling experts to input estimated phase ratios during inference, our method reduces the reliance on expensive labeling methods and leverages expert knowledge to improve segmentation accuracy.
>
> During the inference stage, we mimic expert input by providing phase ratios inputs with 90% accuracy, as detailed in the Appendix A2 section of our paper. We demonstrate that the model achieves better results than the baseline when the expert's phase ratio accuracy is 67% or higher (Figure 9). This indicates that even approximate estimates from experts can significantly enhance model performance and justifies the practical utility of our approach in real-world applications.
>
> **3. Comparison and Evaluation Shortcomings**
>
> **a. Fairness of Comparison**
>
> We appreciate the suggestion to include transformer-based segmentation models like TransUNet and UCTransNet in our comparisons. In response, we have added these models to our experiments, and the results are presented in Table 1 of the revised paper. Our method still demonstrates increased performance, indicating that the inclusion of ratio information provides benefits beyond what attention mechanisms alone can achieve. Additionally, our Spatial Awareness (SA) and Feature Aggregator (FA) modules effectively integrate this additional information, leading to the performance improvements observed in Tables 2 and 3.
>
> **b. Metric Selection**
>
> We agree with the reviewer that dice coefficient may be sufficient but it is a widely used metric in metallographic segmentation studies that allows for direct comparison with prior and future works. Dice coefficient effectively measures the overlap between the predicted segmentation and the ground truth. We do value the input and will be implementing other evaluation metrics in future related studies.
>
> **c. Parameter Comparison**
>
> Thank you for highlighting the importance of comparing model parameters and configurations. We have included Tables 5 and 6 in the appendix of the revised paper. Table 5 compares the performance of different configurations of the SAM model (base, large, and huge), and Table 6 provides a detailed breakdown of the parameters in each of our proposed modules. Our Phase Learning Module, while accounting for approximately 1.2 million additional parameters, improves the performance of the models by over 5% on the private dataset and over 6% on the MetalDAM dataset. This demonstrates that the performance gains are achieved with a reasonable increase in model complexity.

---

> ### Comment · Reviewer_5dQM · 2024-11-26
>
> Thank you for the response. However, I remain unconvinced and still believe this paper does not meet ICLR standard. The main issue is the limited novelty of the AI methodology. The approach is overly simple and intuitive, with no clear justification for why this particular method is necessary or optimal. It lacks the depth and innovation to inspire broader applications.
>
> While I acknowledge the scientific value and the improved experiments from the response, I maintain that the contribution is insufficient for acceptance. My score of 3 reflects that the paper’s average merit does not reach the acceptance threshold.

---

> ### Author Response · Authors · 2024-11-27
>
> We thank the reviewer for their thoughtful review and for sharing their concerns. We appreciate the opportunity to clarify the contributions and significance of our work.
>
> We understand the reviewer’s concern about the perceived limited novelty of our methodology. Our primary goal was to address a significant challenge in microstructure segmentation: **the difficulty of providing effective input conditioning due to the complex and indescribable nature of these images.**
>
>  In existing models like SAM and similar architectures, input conditioning is typically achieved through text prompts or visual cues such as bounding boxes and points. These methods are effective for natural images where objects are distinct and describable. However, in the domain of microstructure images, such as metallographic images, the intricate patterns and lack of intuitive visual cues make it impractical to provide such conditioning inputs. The components within these images often cannot be easily described or annotated with bounding boxes due to their visual complexity and the subtle differences between phases which we describe in Figure 1.
>
> To overcome this challenge, we introduce the use of **phase ratios** as a novel form of input conditioning after extensive thought. Phase ratios represent the proportions of different material phases within a microstructure and can be estimated by experts through visual inspection. By leveraging this readily available expert knowledge, we provide a practical means of conditioning the model without the need for detailed annotations or descriptive prompts.
> Our proposed **PLM** is designed to effectively integrate phase ratio information into the segmentation model. Instead of simply adding the phase ratios as scalar inputs, we transform them into multi-channel ratio maps using the **Ratio Encoder**. These ratio maps are then effectively merged with their corresponding image phases after being spatially aligned.  The input of phase ratios is a single step that can greatly enhance model performance with slight increase in model parameters.
>
> **Justification for the necessity of our method**
>
> The necessity of our method arises from the unique challenges associated with microstructure images:
>
> - **Inadequacy of Traditional Conditioning**: Conventional input conditioning methods are ineffective due to the inability to describe or annotate complex microstructures.
> - **Leverage of Expert Knowledge**: Experts can estimate phase ratios with reasonable accuracy through visual inspection, providing valuable information that is otherwise difficult to encode.
>
> Our method optimally utilizes this expert knowledge by embedding it into the model in a way that enhances segmentation performance without imposing significant additional costs. The integration is seamless and does not require extensive modifications to existing architectures.
>
> **Addressing Concerns About Broader Applications**
>
> While our work focuses on metallographic image segmentation, the underlying concept of using quantitative estimates as conditioning inputs can be extended to other domains facing similar challenges such as medical imaging and remote sensing.
>
> We believe our work addresses a critical gap in the application of deep learning to complex image segmentation tasks where traditional input conditioning fails. By effectively integrating expert-estimated phase ratios through our Phase Learning Module, we provide a novel and practical solution that enhances segmentation performance. Our approach is not only relevant to the materials science community but also offers insights that could inspire broader applications in other fields with similar challenges.

---

### Note · Authors · 2025-01-24

I have read and agree with the venue's withdrawal policy on behalf of myself and my co-authors.